# Osteolytic Lesions in a Sub-Adult Loggerhead Sea Turtle (*Caretta caretta*): A Case Report

**DOI:** 10.3390/ani14091317

**Published:** 2024-04-27

**Authors:** Ignacio Peña Pascucci, Susana Pernas Mozas, Lucía Garrido Sánchez

**Affiliations:** Fundación para la Conservación y Recuperación de Animales Marinos (CRAM), El Prat de Llobregat, 08820 Barcelona, Spain; susana.pernas@cram.org (S.P.M.); lucia.garrido@cram.org (L.G.S.)

**Keywords:** *Caretta caretta*, osteolytic lesions, dysbaric osteonecrosis, osteomyelitis, *Ewingella americana*, decompression sickness, cold-stunning

## Abstract

**Simple Summary:**

Fishery interactions are the most serious conservation risk for sea turtles. The protection of threatened species such as sea turtles is necessary to maintain an ecosystem’s resilience. With a record number of loggerhead sea turtle nests in 2023 in Spain and other Mediterranean countries, the Mediterranean basin is playing an important role in all stages of the loggerhead sea turtle’s life-cycle. Apart from the main cause of admission for rehabilitation of sea turtles, it is necessary to consider the possibility of late-stage affections causing any alteration in the normal physiology or behavior of marine turtles. Osteolytic lesions are a frequent finding during the rehabilitation of sea turtles, and the causes and consequences of them must be studied.

**Abstract:**

Osteolytic lesions in loggerhead sea turtles (*Caretta caretta*) during rehabilitation are attributed to multiple causes, including gas embolism, hypothermia, and osteomyelitis due to bacterial or fungal infection. This study reports the appearance of osteolytic lesions in a sub-adult loggerhead sea turtle with involvement of the right fore and hind flippers, visible swelling of the elbow and knee joints, and accompanied by lameness after 45 days of rehabilitation. Radiographs and computed tomography revealed multiple lytic bone lesions. This was the fourth rehabilitation admission of the turtle after being accidentally captured by trawler ships (bycatch) in 2019, 2020, 2022, and 2023. Potential causes were dysbaric osteonecrosis due to a past decompression sickness event and hypothermia with osteomyelitis from bacterial infection. Blood cultures and antibiotic susceptibility testing led to the isolation of *Ewingella americana* responsive to enrofloxacin. This study investigates extensive fore and hind flipper involvement in a sub-adult loggerhead turtle, aiming to determine causes and risk factors. The pathogenesis and significance of these lesions is discussed.

## 1. Introduction

Human interactions with sea turtle populations are a very frequent problem in coastal areas. Bycatch of non-targeted species is the most serious conservation risk for some marine species like sea turtles [1].

Osteolytic lesions of long bones of the extremities have been documented during the rehabilitation of loggerhead sea turtles. These conditions could be secondary to dehydration, malnutrition, poor perfusion, immunosuppression, and infection [2]. Osteonecrosis as a consequence of ischemia could present an infectious event or not. The possible causes of osteolytic lesions are multiple, and in this case report, the two main differential diagnostics considered were gas embolism causing dysbaric osteonecrosis (DON) and hypothermia with osteomyelitis.

The presence of gas inside the circulatory system is known as gas embolism (GE), and the effect of these bubbles depends on their size and their location [3]. When the normal dive profile of marine animals is modified because of human interaction, adaptations that reduce the accumulation of excess nitrogen in tissue are disrupted. According to Fernandez et al. (2005) [4], gas and fat emboli should be considered in stranded cetaceans suspected of having been exposed to sonar. This issue was also demonstrated in loggerhead sea turtles (*Caretta caretta*) entrapped in trawls and gillnets, demonstrating the development of GE [5].

Decompression sickness (DCS) occurs when there is a sudden decrease in pressure, such as during a rapid ascent from a dive. This rapid pressure change leads to the formation of bubbles, primarily nitrogen, which can result in gas embolism. Blockage of blood vessels and subsequent vessel rupture, or compression of tissues, causes damage to the endothelium and plasma extravasation, triggering clotting and inflammatory responses [6]. A study of gas and fat emboli was well described by Fernandez et al. [4] involving a mass stranding of 10 beaked whales (Family *Ziphiidae*). All the animals in the study had similar pathologic findings of generalized congestion and acute, disseminated hemorrhaging that mainly affected the central nervous system, acoustic jaw fat, pharyngeal and laryngeal serosae, lungs, and kidneys.

According to García-Párraga et al. (2014), turtles submerged in water develop decompression sickness because of heightened activity and the activation of sympathetic responses induced by catecholamines, along with concurrent inhibition of parasympathetic activity. These mechanisms disrupt the usual physiological function of the vagal diving reflex, which normally reduces blood flow through air-filled pressurized lungs during diving.

DON is a form of avascular necrosis affecting the bone, which is seen as a delay consequence of DCS [6]. Considering the typical vascular anatomy around the ends of long bones, it appears plausible that DON is associated with the presence of micro-embolisms containing gas or lipid bubbles. It is suggested that external compression of blood flow in these vessels may occur due to nitrogen bubbles similar to those found in structures surrounding an artery or vein. These compressions, whether within or outside blood vessels, disrupt blood flow in specific areas [7]. Lesions characteristic of DON typically manifest in long bones that house fatty marrow, including the humerus, femur, and tibia. Clinical indications often include joint pain, which may emerge several months or years following exposure to a hyperbaric environment [8].

Avascular necrosis, identified through either distinctive pathological changes in the bone’s articular surfaces or its radiographic presentation in vertebral centra, has been documented in numerous Mesozoic marine reptiles. Its occurrence in both zoological and paleontological records is typically linked with DCS [9]. Furthermore, osteonecrosis-type lesions, which are among the few long-term lesions observable after certain episodes of DCS, have been documented in fossils of mosasaurs and sea turtles dating back to the Cretaceous Age and Late Jurassic periods [10]. According to Moore and Early (2004), the most straightforward explanation for the lesions observed in all examined sperm whale bones from the Atlantic and Pacific oceans over 111 years is that nitrogen emboli caused the osteonecrosis [11].

Another cause of osteolytic lesions observed in sea turtles during the rehabilitation period is related to hypothermia events. Instances of cold-stunning events have been documented among sea turtles in numerous temperate regions of the United States of America and Europe. These events typically occur when water temperatures drop below approximately 15 °C. Osteomyelitis has been recorded during the rehabilitation of cold-stunned Kemp’s ridley (*Lepidochelys kempii*) sea turtles in the United States. This condition is thought to arise secondarily to inadequate blood flow and immune system suppression resulting from hypothermia. In certain cases, stranded turtles with osteomyelitis may exhibit periarticular swelling, abrasions, open wounds, and lameness. In some cases, these symptoms only manifest after several weeks to months of rehabilitation. In turtles with osteomyelitis, osteolytic lesions in the metaphysis and epiphysis of the long bones in the extremities are common. These lesions may persist for many months even after the clinical signs have resolved, posing challenges in determining the appropriate time for releasing the turtle [2].

Typically, the most severe bacterial infections in reptiles, including sea turtles, are caused by Gram-negative aerobic organisms. These bacteria are part of the natural flora found in the oral cavity, skin, and digestive system of reptiles, as well as in the marine environment. Nevertheless, in recent years, infections attributed to Gram-positive bacteria of the genus *Enterococcus* have become more prevalent during the rehabilitation of cold-stunned Kemp’s ridley turtles. *Enterococcus faecalis* has emerged as the most frequently isolated bacteria in cases of septicemia and osteomyelitis in sea turtles at the New England Aquarium [12]. Tsai et al. outline a case involving an olive ridley turtle (*Lepidochelys olivacea*) admitted to a sea turtle rehabilitation center at the National Museum of Marine Biology and Aquarium in Taiwan [13]. The turtle exhibited reduced appetite and swelling in both elbow joints, with *Enterococcus faecalis* being isolated from the affected joints. Also, Geer and collaborators describe a case of mycobacteriosis by *Mycobacterium chelonae* in *Lepidochelys kempii* [14]. The animal, recovered in a rescue center in Baltimore (USA), showed a swollen left elbow joint. Harms et al. report a case of a hypothermic-stunned juvenile Kemp’s ridley sea turtle presenting a right carpal swelling early in rehabilitation [15]. Cytologic examination and bacterial culture of fine-needle aspirate from the carpal joint revealed non-pigmented fungal hyphae and moderate growth of *Nocardia* sp., respectively, diagnosing a mixed unidentified fungal and nocardial osteomyelitis.

In reptiles, osteomyelitis is distinguished by slow, progressive lytic processes, primarily affecting the appendicular skeleton. Local anatomy frequently undergoes alterations due to bone remodeling in response to the infection, often resulting in a persistent lytic defect even after the resolution of the condition [16]. *Aeromona hydrophila*, a Gram-negative bacterium, has been associated with osteomyelitis in a case study of a loggerhead turtle [17].

The current investigation presents a case detailing osteolytic lesions extensively affecting the right fore and hind flippers in a sub-adult loggerhead sea turtle (*Caretta caretta*). The objective of this study was to assess the cause of the lesions, examine risk factors, and describe the physical status of the turtle.

## 2. Materials and Methods

The case corresponded to a bycaught sub-adult (curved carapace length: 62 cm) loggerhead sea turtle, *Caretta caretta,* captured by a trawler ship in January 2023 at La Rápita, Catalonia, Spain. The animal was assisted and transported to the marine animals’ rescue and rehabilitation center CRAM—Fundación para la Conservación y Recuperación de Animales Marinos. CRAM is a non-profit foundation located in El Prat de Llobregat, Barcelona, Spain.

Once at the rescue center, the turtle was identified as CC23/012 and was examined by the veterinarian team. The turtle’s weight and measurements on the first day were 30.5 kg, 57.5 cm straight carapace length (SCL), and 47 cm straight carapace width (SCW). Checking the microchip confirmed that it was the fourth admission of the turtle for rehabilitation at CRAM. The previous rehabilitation periods were in the years 2019, 2020, and 2022.

In March 2019, the same turtle, then identified as CC19/037, was accidentally captured by a trawler ship and presented moderate-to-severe gas embolism (Figure 1), based on the classification proposed by García-Párraga et al. (2014). DCS was diagnosed by radiography and response to hyperbaric chamber treatment. An injury at the level of both fifth lateral (costal) scutes of the carapace (Figure 2) was also present. The treatment for DCS was hyperbaric air at 1.6 atm pressure inside a hyperbaric chamber and 5 mg kg^−1^ enrofloxacin, IM q 48 h for 15 days, with debridement and curettage of the wound. The turtle was released from the beach in May 2019.

In December 2020, CC20/051 suffered a bycatch incident by a trawler ship. At the rescue center, it was diagnosed with pneumonia by radiography (Figure 3) and started treatment with ceftazidime 22 mg kg^−1^, SC q 72 h for 15 days. In January 2021, the turtle was released.

The third admission to rehabilitation was in December 2022 after being accidentally captured by a trawler ship. The turtle presented superficial injuries. After three weeks, at the end of December, the turtle was released.

A month after the third release, in January 2023, the turtle was recaptured once again by a trawler ship. Immediately upon admission, a physical examination was performed. The turtle presented hypothermia (12.9 °C intracloacal temperature) and mentation was assessed as depressed according to Manire et al. 2017, *Sea Turtle Health & Rehabilitation*. Dorsal–ventral (DV), cranial–caudal (CC), and lateral–lateral (LL) radiography was performed, showing no signs of abnormality. Initial treatment included rehydration and a slow increase in body temperature, monitoring room temperature. In the first five hours, the sea turtle’s temperature increased 4 °C, and then the rise of temperature was 4 °C per day until reaching 25 °C, which was the optimal temperature during the rehabilitation period. A solution for rehydration with vitamins was prepared using lactated Ringer’s solution, NaCl 0.9%, and glucose 5% (12 mL kg^−1^ final solution), and it was administrated through the intracelomic route via the inguinal fossa.

Three milliliters of blood sample from the external jugular vein was collected for complete blood count and plasma biochemistry into tubes containing lithium heparin.

Visible swelling and lameness of the right fore and hind flippers were first noted around day 45 of rehabilitation (Figure 4). The turtle had reduced use and range of motion of the affected limbs. Palpation of right elbow and right knee elicited a pain response. Both joints presented increased periarticular soft tissue. A radiographic study of the turtle was performed and showed poorly defined osteolytic process restricted to the distal end of the humerus and femur and the proximal end of the radius and tibia. In the case of DON, changes in imaging are seen early at 4 months after hyperbaric exposure and can be seen as late as 8 months to more than a year [6]. Radiographic studies were repeated every 10–14 days to evaluate the progression of the lesions.

Therefore, to better visualize the lesions, computed tomography (CT) of the whole body, in soft tissue and bone algorithm, with 1.25 mm thick sections, without administration of intravenous iodinated contrast, was performed. CT confirmed the lytic lesions observed in radiography on the distal right humerus, proximal right radius and ulna, distal right femur, and proximal right tibia and fibula. When clinical signs appeared, blood culture at 24–26 °C with antibiogram was performed, revealing a positive result.

The turtle was initially treated with analgesics (Tramadol 5 mg kg^−1^, PO q 48 h), nonsteroidal anti-inflammatory drugs (Meloxicam 0.2 mg kg^−1^, PO q 24 h during one week), and fluid therapy 10 mL kg^−1^ (including 100 mL kg^−1^ of Ringer’s solution, 100 mL kg^−1^ of ClNa 0.9%, and 100 mL kg^−1^ of 5% glucose solution, SC q 24 h for one week). Vitamin and mineral supplements (Stimulfoss; 0.1 mL kg^−1^ with fluids; Aquavits: 3 tablets once a week PO) were also administered. Administration of analgesics and nonsteroidal anti-inflammatory drugs may ease the patient’s pain and increase comfort in everyday activities. Overall, the goal of conservative treatment is to relieve pain, restore a normal range of motion, and delay the progression of the disease [18].

When the results of antimicrobial susceptibility became available, treatment with enrofloxacin (5 mg kg^−1^, PO q 48 h for one month) was initiated.

## 3. Results

Hematologic and biochemical values of blood results for this turtle on the first day of rehabilitation were packed cell volume (PCV) 37% (reference 23 to 34%), hemoglobin 12 g/dL (reference 8 to 14 g/dL), white blood cells (WBC) 25,200/µL (reference 17,000 to 24,000/µL), total protein 5.2 g/dL (reference 1.467 to 6.687 g/dL), calcium 7.4 mg/dL (reference 6.7 to 8.7 mg/dL), phosphorus 4.05 mg/dL (reference 7.773 to 9.073 mg/dL), and aspartate aminotransferase 247 UI/L (reference 44 to 184 UI/L) [19,20].

Swelling, reduced range of motion, and reduced use of the right elbow and right knee were first noted on day 45 of rehabilitation. The turtle was kept in an indoor tank of 6000 m^3^ of sea water during the winter season, and in an outdoor pool of 70,000 m^3^ of sea water during the summer. Physicochemical parameters of water used during the whole rehabilitation period were pH between 7.6 and 7.8, salinity 36 gr/L, average temperature 24.5 °C in winter and 25.5 °C in summer, free chlorine between 0.01 and 0.1 ppm, and total chlorine between 0.1 and 0.2 ppm. Water quality was maintained using a sand filter system, ultraviolet light, and partial marine water changes.

Hard edema with increased periarticular soft tissue and pain was noticed at palpation during clinical evaluation in veterinary medicine [21]. At the beginning of signs, radiographic lesions were polyostotic and poorly defined (Figure 5 and Figure 6).

Early-stage lesions at day 45 of rehabilitation were characterized by a purely osteolytic process restricted to the distal end of the humerus and femur and the proximal end of the radius, ulna, and tibia, including demineralization of the surrounding bones. Osteolytic lesions were restricted to the epiphysis and metaphysis of long bones, and no lesions were detected in the diaphysis.

Late-stage lesions after five months of rehabilitation were characterized by sclerosis and remodeling of the lesion borders (Figure 7). Lesions occurring within the bone shaft typically remain asymptomatic. However, those adjacent to the joint surface, known as juxta-articular lesions, may advance to structural deterioration of the joint surface, leading to symptoms such as pain and restricted movement. CT revealed additional lytic bone lesions of the left ischium, ileum, and pubis; right ischium; left humerus head; and confirmed the lytic lesions observed via radiography (Figure 8). Throughout the treatment period, the animal continued to have a healthy appetite and was bright and alert according to Manire et al. 2017, *Sea Turtle Health & Rehabilitation* [22]. Blood samples were taken once a month to evaluate hematological and biochemical values, and all the parameters were in range according to plasma biochemical and hematological baseline values in loggerhead sea turtles from the Mediterranean Sea proposed by Basile et al. (2011) [19].

On day 45 of rehabilitation, a blood sample for culture was taken and incubated at 24–26 °C for 7 days. *Ewingella americana*, a Gram-negative bacillus, was grown after the incubation period. The antimicrobials tested were amikacin, cefovecin, marbofloxacin, enrofloxacin, gentamicin, doxycycline, clindamycin, amoxicillin with clavulanic, and florfenicol. According to the minimal inhibitory concentration (MIC), the *Ewingella americana* strain was found with high susceptibility to enrofloxacin and marbofloxacin, intermediately susceptibility to doxycycline and florfenicol, and resistance to amikacin, gentamicin, cefovecin, clindamycin, and amoxicillin with clavulanic. At day 15 from the beginning of treatment with enrofloxacin, and 5 days after the end of antibiotic treatment, blood cultures were repeated and, in both cases, yielded no bacterial growth after 7 days at 24–26 °C. A third blood culture was made a month after the last one, with a negative result. At this point, the external appearance of the lesions did not appear to improve.

Before starting antibiotic therapy, fine-needle aspirates for fungal and bacterial culture were taken from the cranial aspect of the right elbow joint following sterile preparation of the overlying skin. Fungal and bacterial cultures were negative.

## 4. Discussion

X-ray is the initial imaging of choice for diagnosis and evaluation of bone lesions. Although radiography showed decalcification and cystic lesions on some bones, it demonstrated a lower sensitivity compared to the CT. One of the hypotheses is that the lesions observed in March 2023 were caused by the decompression sickness event that occurred four years earlier, representing a case of DON. DON is rare but remains extremely important to be recognized as a potential complication of DCS [5,6,11].

When the diving reflex of sea turtles is interrupted by any cause, for example a bycatch accident, the air accumulated in the lungs is absorbed by the systemic circulation. A rapid ascent to the surface without a physiological decompression profile creates gas bubbles mainly composed by nitrogen. Re-compressive hyperbaric oxygen therapy (HBO), which prevents embolism by causing a back-diffusion of gases into the tissues, is the treatment for DCS. Gas that is produced gradually and travels in solution from tissues to capillaries can be gradually expelled by the lungs without forming bubbles [5,7].

It has been suggested that lipid degradation is the beginning of the systemic pathogenic cascades observed in DCS. In fat tissue as opposed to non-fatty tissue, nitrogen is more soluble. Adipocyte size significantly increases with rapid decompression, according to data from human autopsy studies and animal tests [7]. These cells rupture upon additional decompression, releasing their contents to create fat emboli along with other cell lytic detritus. These fat emboli cause ischemic necrosis at the end of long bones, eventually resulting in bone destruction of DON [7]. Fernandez and collaborators (2005) provide the first description of fat emboli in cetaceans, and they have proposed two mechanisms for the formation of fat emboli. Firstly, fat emboli may enter the bloodstream directly after trauma, leading to direct toxic injury in the lungs. This can result in respiratory insufficiency when free fatty acids are released from fat tissues. Secondly, fat emboli can be generated through the disruption of plasma lipoproteins and the coalescence of lipids at the interface of intravascular gas bubbles [4].

The other hypothesis established on this clinic case was that osteolytic lesions were due to hypothermia and osteomyelitis. In January 2023, when the turtle arrived at CRAM, intracloacal temperature was 12.9 °C, diagnosing hypothermia. In addition, the blood culture a week after signs (swelling of joints and lameness of the right fore and right hind flippers) yielded bacillus Gram-negative growth (*Ewingella americana*). Solano et al. (2008) suggest that it remains uncertain whether the osteolytic lesions observed in Kemp’s ridley sea turtles consistently indicate osteomyelitis, or if these lesions could potentially be attributed to sterile necrosis resulting from hypothermia. In mammals afflicted with septic synovitis or osteomyelitis, cultures obtained from blood, synovial fluid, synovial membrane, and bone frequently yield negative results. Diagnosis typically relies on a combination of supportive clinical, laboratory, and imaging findings. The majority of cold-stunned turtles brought in for rehabilitation receive treatment with broad-spectrum antibiotics and antifungal medications due to the high occurrence of sepsis and pneumonia. This treatment is also likely effective in addressing certain osteomyelitis lesions, which could elucidate the clinical improvement observed in these patients, as well as the negative culture results obtained from blood, synovial fluid, and bone in some turtles [2]. As mentioned before, a case of a swollen left elbow joint by *Mycobacterium chelonae* was well described by Greer et al. in *Lepidochelys kempii* [14]. Pace and collaborators (2018) have been associated a case of polyostotic osteomyelitis in a loggerhead sea turtle with *Aeromona* spp. [17,23].

Harms et al. (2002) documented a case of a hypothermic-stunned juvenile Kemp’s ridley sea turtle exhibiting mixed fungal and nocardial osteomyelitis in the right carpal region [15]. Several aspects of the pathogenesis of osteolytic lesions in cold-stunned sea turtles remain uncertain. These lesions could develop secondarily to immunosuppression and the presence of bacteremia or fungemia, with microbial seeding of devitalized bone following ischemia induced by hypothermia [24].

In sea turtles, the primary routes for bacterial infection typically involve traumatic injuries and the aspiration of water. Bacteria can enter the bloodstream through these routes and spread throughout the entire body [17]. In this case, it is possible that the loggerhead turtle developed osteomyelitis as a consequence of *Ewingella americana* septicemia from an initial site that could not be established with certainty. There are some reports in humans of osteomyelitis caused by *Ewingella americana* [25,26]. In the first and the second rehabilitation period, the turtle presented a severe wound in the carapace and radiographic signs compatible with pneumonia, respectively. In the first case, the turtle was treated with enrofloxacin, and in the second one with ceftazidime. The hematogenous spread of bacteria has been frequently cited as the underlying cause of osteoarthritis and septic arthritis in sea turtles [27].

## 5. Conclusions

Osteolytic lesions are commonly observed in sea turtles during rehabilitation. Sometimes, those lesions show radiological findings without clinical signs that can compromise the turtle’s health.

Assessing the specific event that caused osteonecrosis in long bones of a sea turtle that arrives for rehabilitation may be difficult, and it is probable there would be more than one cause related to the development of the lesion. The first step recommended would be to try to determine if we are dealing with septic or aseptic osteonecrosis. With this aim, blood and joint fluid culture, cytologic, and histopathologic examination should be necessary to determinate the etiology of the lesion. Also, avascular osteolytic lesions secondary to an ischemic event could develop after gas embolism or hypothermia.

Additional imaging techniques such as CT scans should be used for early diagnosis of dysbaric osteonecrosis when an X-ray does not reveal any lesions, especially for those animals who have a history of decompression sickness and/or joint symptoms. Clinicians should be aware of the possibility of osteolytic lesions for hypothermia, osteomyelitis, and DON in rehabilitation of sea turtles and should pursue additional diagnostic testing [24]. Given the high frequency of sea turtles diagnosed with gas embolism after bycatch over recent years, DON should be considered in differential diagnosis of osteolytic lesions in marine turtles.

Furthermore, more importantly, better-equipped diagnostic rehabilitation labs, able to run the various tests to clarify causes for sickness, and specialized sea turtle veterinary care side by side with conservation research, are needed to safeguard these vulnerable and legally protected species.

## Figures and Tables

**Figure 1 animals-14-01317-f001:**
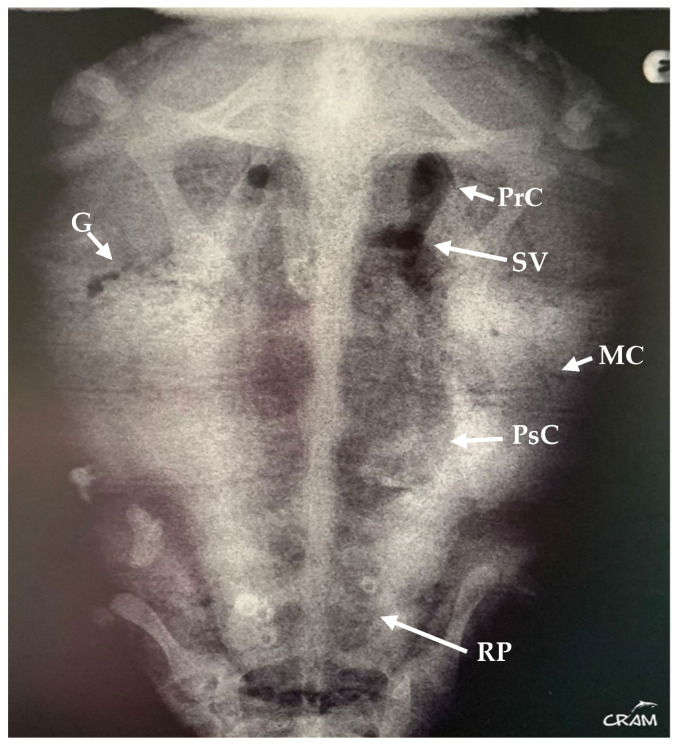
Dorsoventral radiograph of CC19/037 with GE of moderate-to-severe degree. Gas is evident in precava (PrC) and postcava (PsC) veins, sinus venosus (SV), gastric vessels (G), marginocostal vessels (MC), and renal portal vessels (RP).

**Figure 2 animals-14-01317-f002:**
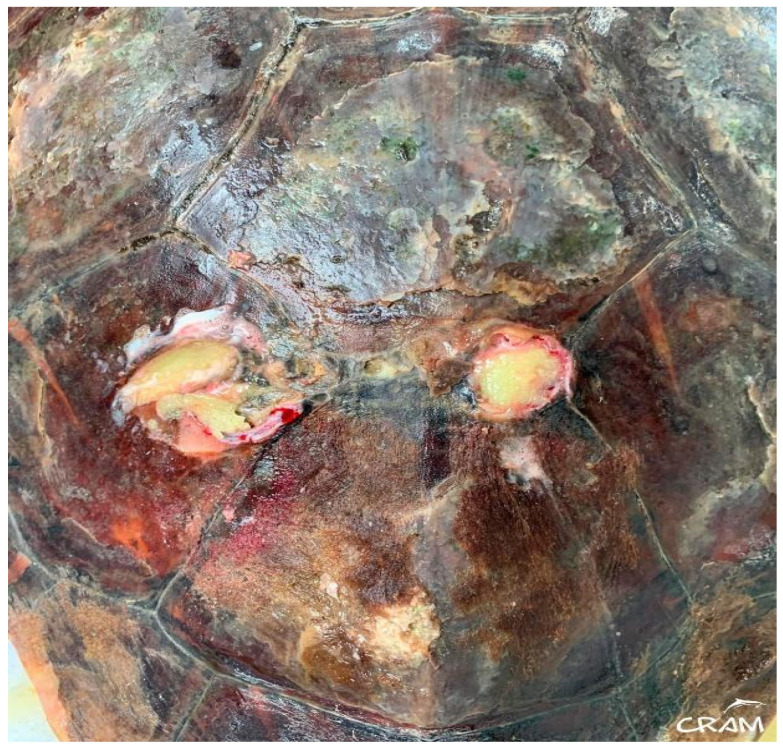
Appearance of the wound.

**Figure 3 animals-14-01317-f003:**
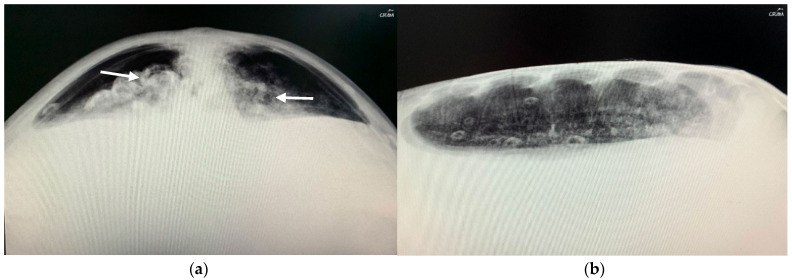
Radiographic examinations in cranio-caudal (**a**) and lateral (**b**) projections of the lungs. The images show an opacification of the whole parenchyma following bacterial pneumonia. The arrows show a densification of the peribronchial lung parenchyma.

**Figure 4 animals-14-01317-f004:**
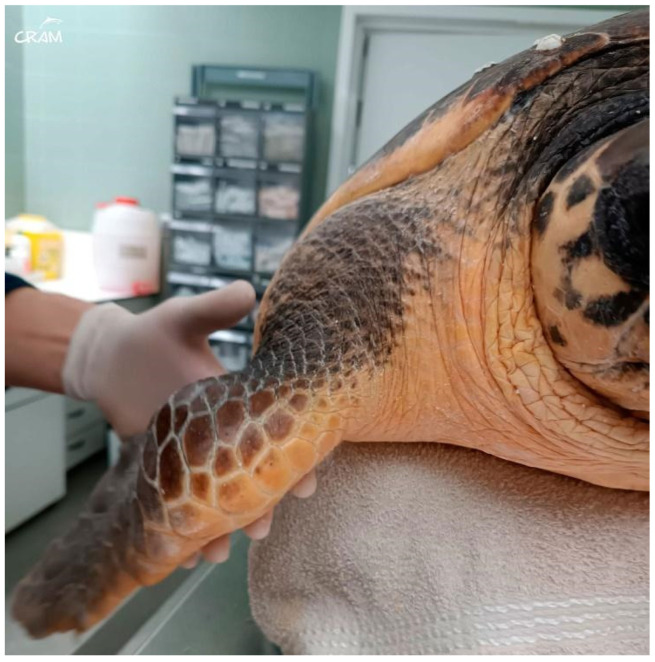
Visible swelling of the right front flipper at the elbow joint.

**Figure 5 animals-14-01317-f005:**
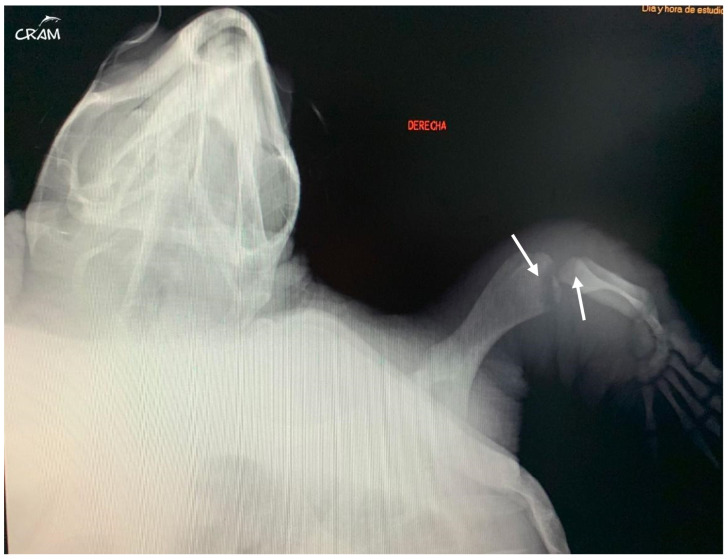
Dorsoventral radiograph of the right front flipper at day 45 of rehabilitation. Well-defined bone lesions (arrows) are noted. Lesions are lytic with relevant osteopenia and no bone remodeling. The lytic process is seen communicating with the joint space in the elbow joint.

**Figure 6 animals-14-01317-f006:**
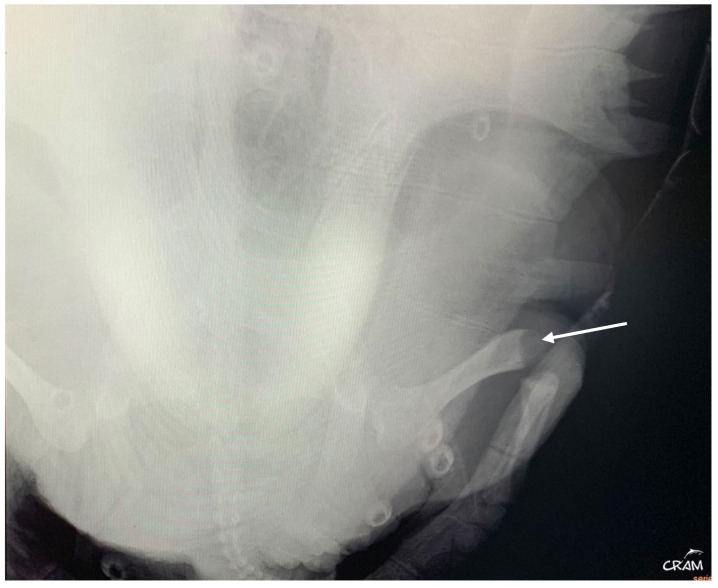
Right pelvic girdle in dorso-ventral view at day 45 of rehabilitation. Visible bone lysis on the distal end of the right femur (arrow).

**Figure 7 animals-14-01317-f007:**
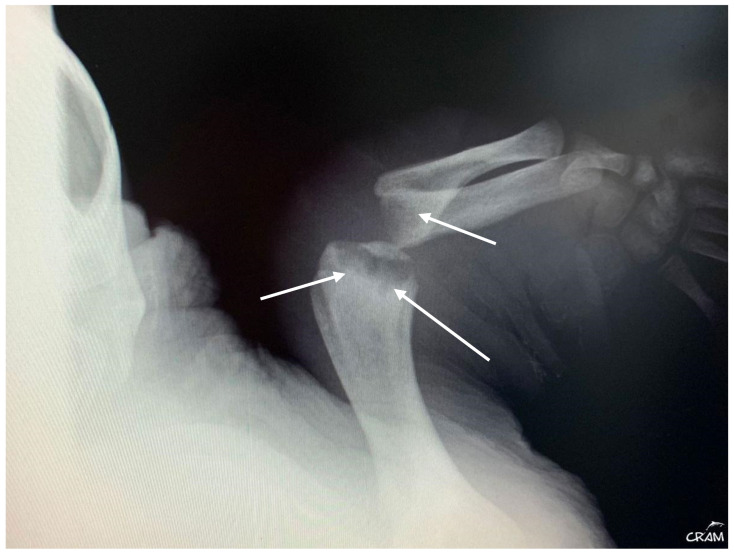
Dorsoventral radiographic of the right flipper after five months of rehabilitation. Well-defined sclerosis surrounding rounded lesion borders is noted (arrows).

**Figure 8 animals-14-01317-f008:**
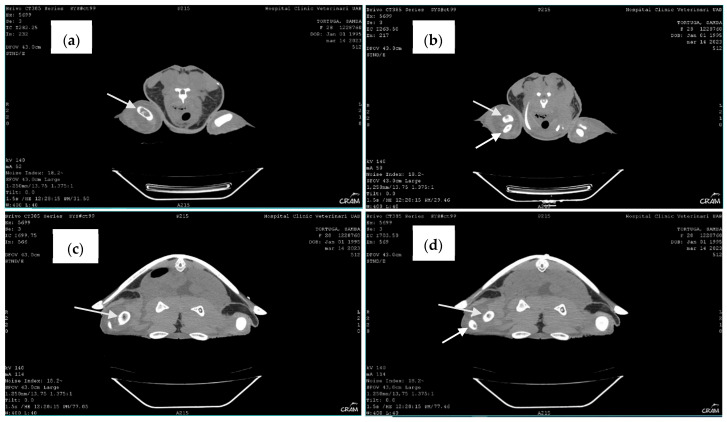
Computed tomography. Note bone lysis on the distal end of the right humerus (arrow) (**a**), the proximal end of the right radius and ulna (arrows) (**b**), and the proximal end of the right tibia and fibula (arrows) (**c**,**d**).

## Data Availability

No new data were created or analyzed in this study. Data sharing is not applicable to this article.

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
