# Peer review of "Osteolytic Lesions in a Sub-Adult Loggerhead Sea Turtle (Caretta caretta): A Case Report"

_animals, 2024, doi:10.3390/ani14091317_

Round 1

Reviewer 1 Report

Comments and Suggestions for Authors

This work may be improved and made to focus further on sea turtles and Caretta caretta in particular.  

Author Response

Thank you very much for taking the time to review this manuscript. Please find the detailed responses below. Please see the attachment

Comments 1: What about reviews on bacterial infections on sea turtles such as that by Ebani 2023 which also includes a case of swollen elbow in a sea turtle.

Also what about other possible causes including fungal osteomyelitis...refer to Harms et al. 2002.

Response 1Tsai et al. [13] describe a case in an olive ridley turtle (Lepidochelys olivacea) admitted to a sea turtle rehabilitation facility at the National Museum of Marine Biology and Aquarium in Taiwan, showing lack of appetite and bilateral elbow joint swelling, where Enterococcus faecalis was isolated from the affected joints. Also, Geer and collaborators [14] describe a case of mycobacteriosis by Mycobacterium chelonae in Lepidochelys kempii. The animal, recovered in a rescue center in Baltimore (USA), showed a swollen left elbow joint. Harms et al. [15] report a case of a hypothermic-stunned juvenile Kemp’s ridley sea turtle presenting a right carpal swelling early in rehabilitation. Cytologic examination and bacterial culture of fine needle aspirate from the carpal joint, revealed non-pigmented fungal hyphae and moderate growth of Nocardia sp., respectively, diagnosing a mixed unidentified fungal and nocardial osteomyelitis.

Comments 2: Pace et al. 2018 particularly focused on sea turtle Aeromonas induced polyostotic ostemomelitis. Therefore other refs are required in reference to the publications on snakes and lizards.
Also no clear mention of the possible cause for osteomylitis as indicated by Pace et al. 2018

Response 2In reptiles, osteomyelitis is characterized by both slow progressive lytic processes, predominantly in the appendicular skeleton. Local anatomy is often altered by bone remodeling in response to infection, and a persistent lytic defect is often seen after the resolution of the process [16]. Aeromona hydrophila, a gram-negative bacteria, have been associated with osteomyelitis in a case study of a loggerhead turtle [17].

Comments 3 and 4: Details of trawler ship: gear, target fishing organisms, death, working at night or by day.

Any GPS available? if yes include for scientific record of where the incident occurred.

Response 3: Unfortunately, we don't have that information

Comments 5: Any identification of the bacteria causing the pneumonia? Was it Chlamydia pneumoniae?

Response 5: Unfortunately, we don't have that information.

Comments 6: Indicate which reference is being used to rate mentation.  Is it Chrisman et al in 1997 or others?

Response 5: The turtle presented hypothermia (12.9 ºC intracloacal temperature) and mentation was assessed as depressed according to Manire et al 2017, Sea Turtle Health & Rehabilitation. 

Comments 7: at what rate and up to what temperature?

Response 7: The first five hours, sea turtle temperature increased 4°C, and then the rise of temperature was 4°C per day until 25°C, which was the optimal temperature during the rehabilitation period.

Comments 8: How many mls of blood was sampled and what were the blood tests planned?

Response 8: Three millilitres of blood sample from the external jugular vein were collected for complete blood count and plasma biochemistry into tubes containing lithium heparin.

Comments 9: Ideally for all parameters you should include some published works confirming these reference values for turtles in the Mediterranean for the age of your turtle specimen.

Response 9: Hematologic and biochemical values of blood results for this turtle the first day of rehabilitation were: packed cell volume (PCV) 37% (reference 23 to 34%), hemoglobin 12 g/dl (reference 8 to 14 g/dl), white blood cells (WBC) 25200/µL (reference 17000 to 24000/µL), total protein 5.2 g/dL (reference 1.467 to 6.687 g/dl), Calcium 7.4 mg/dl (reference 6.7 to 8.7 mg/dl), Phosphorus 4,05 mg/dl (reference 7.773 to 9.073 mg/dl) and Aspartate aminotransferase 247 UI/L (reference 44 to 184 UI/L) [19, 20].  

Comments 10:  Would be important to describe the conditions of the rehabilitation at some point as well.

Response 10: The turtle was maintaining in an indoor tank of 6.000 m3 of sea water during the winter season, and in an outdoor pool of 70.000 m3 of sea water during the summer. Physicochemical parameters of water used during the whole rehabilitation period were Ph between 7.6-7.8, salinity 36 gr/L, average temperature 24.5 °C in winter and 25.5 °C in summer, free chlorine between 0.01-0.1 ppm and total chlorine between 0.1-0.2 ppm. Water quality was maintaining using a sand filter system, ultraviolet light and partial marine water changes.

Comments 11: Indicate at what day of rehabilitation these radiographs (Fig 5 & 6) were taken.

Response 11: At day 45 of rehabilitation

Comments 12: At what stage during the rehabilitation?

Response 12: at day 45 of rehabilitation 

Comments 13: At what stage during the rehabilitation?

Response 13: after five months of rehabilitation 

Comments 14: What parameters were used to assess these conditions?

Response 14: Throughout the treatment period, the animal continued to have a healthy appetite and was bright and alert according to Manire et al 2017, Sea Turtle Health & Rehabilitation. Blood samples were taken once a month to evaluate hematological and biochemical values, and all the parameters were in range according to plasma biochemical and hematological baseline values in loggerhead sea turtles from the Mediterranean Sea proposed by Basile et al. (2011) [19].

Comments 15: Describe the changes in the appearance and conditions of the sea turtle's appendiges/lesions after infection was stopped.

Response 15: At this point, the external appearance of the lesions didn´t appear to improve.

Comments 16: perhaps include references 4 and 10 too which are more relevant to marine species?

Response 16: I agree

Comments 17: Here again reference and comparison with other works such as that of ref 4 and others may be better as they focus on marine turtles.

Response 17: I agree. Fernandez and collaborators (2005) provide the first description of fat emboli in cetaceans. Two mechanisms have been proposed by Fernandez et al. for the development of fat emboli. First, direct entry of fat emboli into the bloodstream after trauma may cause direct, toxic injury in the lung and produce respiratory insufficiency when free fatty acids are released from fat tissues. A second mechanism involves the generation of fat emboli from plasma lipoprotein disruption and coalescence of lipid at the intravascular gas bubble interface [4].

Comments 18: Expand on the whole range of bacterial and fungal infections and other parameters tested for in cold stunned sea turtles. Ref to: Ebani et al 2023, Pace et al 2018, innis CJ et al 2014, Harms et al. 2002, George RH 1997, Spotia JR et al 1997, Niemuth JN et al 2020, etc. 

Response 18: As mentioned before, a case of a swollen left elbow joint by Mycobacterium chelonae was well described by Greer et al. [14] in Lepidochelys kempii. Pace and collaborators (2018) have been associated a case of polyostotic osteomylitis in a loggerhead sea turtle with Aeromona spp. [17, 21].

Harms et al. (2002) report a hypothermic stunned juvenile Kemp´s ridley sea turtle with right carpal mixed fungal and nocardial osteomyelitis [15]. Some aspects of the pathogenesis of osteolytic lesions in cold-stunned sea turtles remain unclear. Lesions may occur secondary to immunosuppression and bacteremia or fungemia, with microbial seeding of devitalized bone subsequent to ischemia from hypothermia [22].

Comments 19: what about references that have considered MRI in turtles such as Valente et al 2006....and Ana Luisa Schifino Valente PhD 2007. or animals by Bowen A., et al., 2022

Response 19: Thank you for pointing this out

Comments 20: add reference/s where this recommendation has already been supplied in literature.

Response 20: CT scan should be used for early diagnosis of dysbaric osteonecrosis when X-ray does not reveal any lesions, especially for those animals who had a history of decompression sickness and/or joint symptoms. Clinicians should be aware of the possibility of osteolytic lesions for hypothermia, osteomyelitis and DON in rehabilitation of sea turtles and should pursue additional diagnostic testing [22]. 

Comments 21: clarify what is meant by "affections"

Response 21: DON

Comments 22: More importantly better equipped diagnostic rehabilitation labs, able to run the various tests to clarify causes for sickness, and specialized sea turtle veterinary care side by side with conservation research are needed to safeguard these vulnerable and legally protected species.

Response 22: Thank you for pointing this out.

Reviewer 2 Report

Comments and Suggestions for Authors

The paper submitted by the authors is quite interesting and provides additional information of complications produced by DCS. Nonetheless, there are issues that should be addressed before acceptance.

Specific comments 

Line 10, and osteomyelitis due to… Line 10, this study reports (the diseases) on… I recommend to rewrite the abstract section in order to clarify the specific information. The first paragraph of the introduction section shows some redundant sentences, please check.

Line 44 or line 52, here add these references since they were pivotal in the diagnostic of DCS in marine mammals  (Fernandez et Al., 2005, https://pubmed.ncbi.nlm.nih.gov/16006604/)  

Line 114, what criteria did you use to this classification?

Line 129, since you are adding information of the hyperbaric chamber, you should add the same about the Ab, dosage and time. Same with line 178. Lines 206-207, move this sentence to the discussion section.

Line 216, here you jump to CT study without any explanation. Therefore, it is important to add some sentences as “ to better visualize the lesions (extension…). Here, you should add how was the CT attenuation here.

Lines 218-219, please move them to the discussion section.

Line 220, please add some information about the antibiogram results.

Lines 234-235, please move this sentence to discussion section, adding a reference. Figure 5, there is quite relevant osteopenia that is not mentioned here. Figure 8, please add arrows pointing the lesions, and specifying the bones affected.

Line 301, please specify the day of the cytology. It is important to highlight the findings observed in the cytology when the bacteria were identified. In addition, why don’t you mention about possible septic arthritis?

Lines 313-316, this paragraph is redundant, please delete.

Lines 325-327, this paragraph should be supported by a reference. In addition, it does not sound clear.

Lines 385-387, however, it seems that the authors do not mention this disease in the results section. The conclusion is quite long and should be shortened.

Author Response

Thank you very much for taking the time to review this manuscript. Please find the detailed responses below. Please see the attachment.

Comments 1: Line 10, and osteomyelitis due to… Line 10, this study reports (the diseases) on… I recommend to rewrite the abstract section in order to clarify the specific information. The first paragraph of the introduction section shows some redundant sentences, please check.

Response 1: I agree

Comments 2: Line 44 or line 52, here add these references since they were pivotal in the diagnostic of DCS in marine mammals  (Fernandez et Al., 2005, https://pubmed.ncbi.nlm.nih.gov/16006604/)  

Response 2: When the normal dive profile of marine animals is modified because of human interaction, adaptations that mitigate the accumulation of excess nitrogen in tissue, are disrupted. According to Fernandez et al. (2005) [4], gas and fat emboli should be considered in stranded cetaceans suspected of having been exposed to sonar. 

Comments 3: Line 114, what criteria did you use to this classification?

Response 3: curved carapace length: 62 cm

Comments 4: Line 129, since you are adding information of the hyperbaric chamber, you should add the same about the Ab, dosage and time. Same with line 178. Lines 206-207, move this sentence to the discussion section

Response 3: 5 mg kg-1 enrofloxacin, IM q 48 hr for 15 days.

ceftazidime 22mg Kg-1, SC q 72 hr for 15 days. 

Comments 5: Line 216, here you jump to CT study without any explanation. Therefore, it is important to add some sentences as “ to better visualize the lesions (extension…). Here, you should add how was the CT attenuation here

Response 5: Agree. Therefore, to better visualize the lesions a computed tomography (CT) of the whole body, in soft tissue and bone algorithm, with 1.25mm thick sections, without administration of intravenous iodinated contrast, was done. 

Comments 6: Lines 218-219, please move them to the discussion section

Response 6: I agree

Comments 7: Line 220, please add some information about the antibiogram results

Response 7: Lines 325-330

Comments 8: Lines 234-235, please move this sentence to discussion section, adding a reference. Figure 5, there is quite relevant osteopenia that is not mentioned here. Figure 8, please add arrows pointing the lesions, and specifying the bones affected.

Response 8: I agree

Comments 9:  Line 301, please specify the day of the cytology. It is important to highlight the findings observed in the cytology when the bacteria were identified. In addition, why don’t you mention about possible septic arthritis?

Response 9: On March 14, 2023, fine needle aspirates for fungal and bacterial culture were taken from the cranial aspect of the right elbow joint following sterile preparation of the overlying skin. Fungal and bacterial cultures were negative.

Comments 10: Lines 313-316, this paragraph is redundant, please delete

Response 10: I agree

Comments 11: Lines 325-327, this paragraph should be supported by a reference. In addition, it does not sound clear

Response 11: Thank you for pointing this out

Comments 12: Lines 385-387, however, it seems that the authors do not mention this disease in the results section. The conclusion is quite long and should be shortened

Response 12: I Agree.

Round 2

Reviewer 2 Report

Comments and Suggestions for Authors

The revised version provided by the authors shows important improvement and can be accepted after minor changes, which are mentioned below.

Specific comments

Line 66, start this sentence as follows " this issue was also demonstrated in loggerheads sea turtles entrapped in trawls and gillnets, demonstrating...

In discussion section, please join the first and second paragraph. In addition, in line 547, after “evaluation”, please add “ in veterinary medicine” and put some references.   In the conclusion,  Line 701, please change as follows “Sometimes, those lesions show radiological findings without clinical signs that can compromise turtle’s health” Line 703, change “finding” by “assessing”. Line 704, please add “,” after difficult. Line 705, please remove “,” after recommend. Line 710, please add “additional imaging techniques such as CT… Line 717, add “Furthermore” Line 719, please add “,” after “research”

Author Response

Thank you very much for your help and your corrections. All the minor changes were made. Please see the attachment with the corrections in red.
